# Infrared HOT Photodetectors: Status and Outlook

**DOI:** 10.3390/s23177564

**Published:** 2023-08-31

**Authors:** Antoni Rogalski, Małgorzata Kopytko, Weida Hu, Piotr Martyniuk

**Affiliations:** 1Institute of Applied Physics, Military University of Technology, Kaliskiego 2, 00-908 Warsaw, Poland; antoni.rogalski@wat.edu.pl (A.R.); malgorzata.kopytko@wat.edu.pl (M.K.); 2State Key Laboratory of Infrared Physics, Shanghai Institute of Technical Physics, Chinese Academy of Sciences, 500 Yu-Tian Road, Shanghai 200083, China; wdhu@mail.sitp.ac.cn

**Keywords:** HOT IR detectors, HgCdTe photodiodes, BLIP performance, 2D material photodetectors, cascade photodetectors, colloidal quantum dot photodetectors

## Abstract

At the current stage of long-wavelength infrared (LWIR) detector technology development, the only commercially available detectors that operate at room temperature are thermal detectors. However, the efficiency of thermal detectors is modest: they exhibit a slow response time and are not very useful for multispectral detection. On the other hand, in order to reach better performance (higher detectivity, better response speed, and multispectral response), infrared (IR) photon detectors are used, requiring cryogenic cooling. This is a major obstacle to the wider use of IR technology. For this reason, significant efforts have been taken to increase the operating temperature, such as size, weight and power consumption (SWaP) reductions, resulting in lower IR system costs. Currently, efforts are aimed at developing photon-based infrared detectors, with performance being limited by background radiation noise. These requirements are formalized in the Law 19 standard for P-i-N HgCdTe photodiodes. In addition to typical semiconductor materials such as HgCdTe and type-II A^III^B^V^ superlattices, new generations of materials (two-dimensional (2D) materials and colloidal quantum dots (CQDs)) distinguished by the physical properties required for infrared detection are being considered for future high-operating-temperature (HOT) IR devices. Based on the dark current density, responsivity and detectivity considerations, an attempt is made to determine the development of a next-gen IR photodetector in the near future.

## 1. Introduction

The dominant position in infrared (IR) photon detector technology is still held by HgCdTe. This material system has inspired the development of three “generations” of devices for both military and civilian use. The need for cooling significantly limits the widespread application of this technology. A milestone in the development of room-temperature IR cameras was the invention of the microbolometer array. However, they belong to the family of thermal detectors that have a limited time response—usually in the millisecond range—and are not very useful for applications requiring multispectral detection. For these reasons, intensive efforts are being made to develop IR systems based on photon detectors and to increase their operating temperature by reducing their size, weight and power consumption (SWaP). The literature refers to this class of photodetectors as high-operating-temperature (HOT) devices [1], i.e., operating at a temperature close to room temperature.

The goal of this paper is to forecast the development of IR photon detectors in the next decade, operating at room temperature. For this reason, various material systems used in IR detectors are considered. Special attention is paid to those photodetectors with performance being determined by the influence of background radiation noise (BLIP limitation)—fully depleted P-i-N HgCdTe photodiodes, photodetectors made of 2D materials and CQDs.

## 2. Figure of Merit for IR Detector Materials

Normalized detectivity (*D**) is a convenient parameter for the comparison of IR photodetectors. As demonstrated by Piotrowski and Rogalski [2], it is limited by the statistical nature of the generation and recombination mechanisms in a semiconductor and is expressed by the formula
(1)$$D^{*}=k\frac{\lambda }{hc}{\left(\frac{\alpha }{G_{th}}\right)}^{1/2}$$
where *λ* is the wavelength, *h* is Planck’s constant, *c* is the velocity of light, *α* is the absorption coefficient, *G_th_* is the thermal generation in the active volume (in cm^−3^s^−1^) and *k* is the coefficient depending on the radiation coupling, due to, e.g., antireflection coatings, microcavities or plasmonic structures. The figure of merit of any IR detector material that can be utilized to predict its ultimate performance is the ratio of the absorption coefficient to the thermal generation rate, α/*G_th_*. It is also useful in selecting the material for the active area of the detector.

After short transformations, it can be shown that the detectivity is proportional to the product of $\sqrt{\alpha \tau }$ [3]
(2)$$D^{*}\propto {\left(\frac{\alpha }{G_{th}}\right)}^{1/2}={\left(\frac{\alpha N_{maj}\tau }{n_{i}^{2}}\right)}^{1/2}=\left(\frac{\sqrt{N_{maj}}}{n_{i}}\right)\sqrt{\alpha \tau }$$
where *N_maj_* is the majority carrier concentration, *n_i_* is the intrinsic carrier concentration and *τ* is the lifetime of minority carriers.

Figure 1 shows the absorption coefficient versus bandgap energy for different material systems at room temperature. The absorption coefficient was estimated for the threshold energy of 1.2 × *E_g_*, where *E_g_* is the semiconductor bandgap. For HgCdTe, the experimental data were taken from the work by Chu et al. [4]. The relationships presented for superlattices should be treated as approximate, due to some literature discrepancies. In general, the absorption coefficient for a superlattice is lower than for HgCdTe for a given bandgap energy. Moreover, the absorption coefficient of an InAs/InAsSb type-II superlattice (T2SL) is approximately half that of an InAs/GaSb superlattice at *λ* = 8 μm [5]. The same ratio was adopted for other wavelengths. Numerical modeling by Klipstein et al. [6] also confirmed the weaker coefficient of Ga-free superlattices.

Two-dimensional materials are derived from layered solids, in which atoms are linked together by ionic or tight covalent bonds in the 2D plane, while each of these atomic planes is bonded together by weak van der Waals (vdW) interactions along the out-of-plane direction. This atomic arrangement means that many 2D materials can be mechanically exfoliated from layered vdW solids. Furthermore, due to the weak physical bonds between individual atomic planes, it is possible to combine different 2D materials, along with the possibility of the arbitrary formation of heterostructures.

Although layered materials are made of the same atoms as their bulk counterparts, their band structures differ from each other. For example, in the case of transition metal dichalcogenides (TMDs), such as MoS_2_, MoSe_2_, WS_2_ and WSe_2_, the bandgap can be tuned by varying the number of layers—from a smaller indirect transition to a larger direct transition due to quantum confinement effects. Therefore, thanks to the tunable bandgap, TMDs can detect light of different wavelengths. In addition, the optical and electronic properties of these materials are strongly modified by the large strains.

Based on the data collected in Ref. [8], the most important conclusions regarding the absorption coefficients of TMDs and HgCdTe are as follows:the absorption coefficients of HgCdTe (*E_g_* ≈ 0.1–0.3 eV) are below 10^4^ cm^−1^ and for TMDs (*E_g_* > 1 eV) above 10^5^ cm^−1^, andfor a hypothetical TMD with an energy gap of 0.1–0.2 eV, the abortion coefficient can be expected to be below 10^5^ cm^−1^.

Figure 2 shows the minority carrier lifetime versus cut-off wavelength for different material systems at room temperature [9,10,11]. Among the presented materials, HgCdTe is characterized by favorable internal recombination mechanisms, which determine the long carrier lifetime and favor the operation of photodetectors in HOT conditions. The carrier lifetime of a lightly doped HgCdTe (~10^13^ cm^−3^) is specified by the Shockley–Read–Hall (SRH) mechanism. As is shown in Ref. [12], the highest reported values are 10 ms and 0.5 ms for a mid-wave infrared (MWIR) and a long-wave infrared (LWIR) absorber, respectively. These values were determined from the photodiode dark currents at low temperatures. Estimated lifetimes can be taken for photodetectors operating at room temperature [11]. Nevertheless, it should be noted that at 300 K, due to the high intrinsic carrier concentration (6 × 10^15^ cm^−3^ at MWIR, and 5 × 10^16^ cm^−3^ at LWIR range), the minority carrier lifetime is limited by Auger mechanisms—Auger 1 in the n-type and Auger 7 in the p-type material. The typical relation between intrinsic Auger 1 and Auger 7 lifetimes in HgCdTe is *τ*_Ai7_ ≈ 6*τ*_Ai1_ [11].

An unknown mechanism creates SRH centers in semiconductors, involving residual impurities and native defects. In general, A^III^B^V^ compounds exhibit more active SRH centers in comparison with A^II^B^VI^ ones including HgCdTe, resulting in lower minority carrier lifetimes. For example, the InAs/GaSb T2SL carrier lifetime limited by the SRH mechanism is typically several tens of nanoseconds, both in MWIR and LWIR absorbers. It is believed that the short lifetime is due to the presence of a shallow state near the GaSb valence band, which falls directly within the bandgap of InAs/GaSb SLSs. Ga-free InAs/InAsSb superlattices exhibit longer lifetimes, up to several microseconds for the MWIR region. In Ref. [13], the SRH lifetimes were determined from dark currents for the best InAs/InAsSb T2SL detectors. The obtained value for the MWIR absorber was 25 μs, and that for the LWIR absorber was 5 μs.

However, compared to a low-doped HgCdTe, SRH carrier lifetimes of InAs/InAsSb T2SL are up to three orders of magnitude lower for a similar bandgap. It is presumed that the SRH recombination mechanism is related to some semiconductors’ deviation from ideal crystallinity. In A^II^B^VI^ alloys, the ionic bonds are stronger than in the corresponding A^III^B^V^ materials. For this reason, the electron wave function around the lattice sites is much more strongly compact, making the crystal lattices of A^II^B^VI^ compounds less prone to the formation of band states due to crystal imperfection [11].

Table 1 presents the estimated $\sqrt{\alpha \tau }$ figures of merit for HgCdTe, InAs/GaSb and Ga-free InAs/InAsSb superlattices. It should be noted here that at 300 K, the highest reported values of the carrier lifetimes limited by the SRH mechanism can be obtained only with the appropriate detector design, in which the intrinsically generated carriers will be reduced to at least the doping level, or, as in the case of the HgCdTe P-i-N photodiode, to the complete depletion of the absorber. Between the two types of T2SLs, the difference in the $\sqrt{\alpha \tau }$ values is an order of magnitude at LWIR and two orders of magnitude at the MWIR range. A better superlattice system has still a lower $\sqrt{\alpha \tau }$ value compared to HgCdTe. This difference is mainly due to the much longer carrier lifetime in HgCdTe, as discussed above.

Regarding Figure 2, the demonstrated carrier lifetime in two-dimensional materials is at the nanosecond level and is approximately five orders of magnitude lower than for the HgCdTe ternary alloy. The dominant recombination pathway for TMDs (MoS_2_, MoSe_2_ and WSe_2_) is exciton recombination [14,15,16]. Considering the experimental data collected above for LWIR HgCdTe (*α* = 2.2 × 10^3^ cm^−1^, *τ* = 0.5 ms) and TMDs (*α* = 2 × 10^5^ cm^−1^, *τ* = 1 ns), it is also interesting to compare the estimated $\sqrt{\alpha \tau }$ values for these two material systems. For HgCdTe, $\sqrt{\alpha \tau }$ is equal to 1.05 (s/cm)^1/2^, while, for TMD materials, it is 1.4 × 10^−2^ (s/cm)^1/2^, which is two orders of magnitude smaller. However, this is a value close to the one predicted for InAs/InAsSb T2SL. It should be noted that the value of $\sqrt{\alpha \tau }$ was compared for semiconductors with significantly different bandgaps. The absorption coefficient for TMDs was taken for an energy gap on the order of 1 eV, while, for HgCdTe, it was 0.1 eV. For a hypothetical 2D material with a narrow energy gap (0.1 eV, as is the case of HgCdTe), the absorption coefficient should be smaller (by approximately one order of magnitude), which would result in an even smaller $\sqrt{\alpha \tau }$ value compared to HgCdTe.

Considering electron mobilities, TMD 2D materials do not show a distinct advantage over the conventional 3D bulk materials. Figure 3 compares the electron mobilities for selected materials, namely group-6 TMDCs, black phosphorus (bP) and typical noble TMDs on a back-gated SiO_2_ substrate, with the standard bulk semiconductors used in IR detector (HgCdTe and A^III^B^V^ alloys) production.

## 3. Ultimate HgCdTe Photodiode Performance

For more than 60 years, since 1959, when HgCdTe was discovered [18], this material system has successfully overcome the major challenges of different types of detector families, including extrinsic silicon, Schottky barriers on silicon, lead–tin telluride devices, AlGaAs multiple quantum wells, T2SL and especially silicon microbolometers. This leads to the following question: will the further development of 2D materials affect the privileged position of HgCdTe?

In the literature, it has been accepted to compare the performance of different types of IR detectors with developed rules for HgCdTe, such as Rule 07 [19] and Rule 22 (an update of Rule 07) [20], as well as Law 19 [12]. Thus far, Rule 07 has become very common among the IR community as a reference technology. Since the first two rules are empirically derived standards for p-on-n HgCdTe photodiodes constrained by the Auger 1 mechanism due to an extrinsic doping concentration in the active region of ~10^15^ cm^−3^ for Rule 07 and <10^15^ cm^−3^ for Rule 22, it is expected that for a very low-doping active region (around 10^13^ cm^−3^), the values of the photodiode dark current density can be lower. However, this is not the case for Law 19 due to the fact that it is a relationship determined by a fundamental law—the dark current density is limited by the background radiation from the surrounding environment reaching the detector. This current is crucial because it is not due to imperfections in the material, detector design or associated electronics, but rather to the detection process itself, which is determined by the discrete nature of the radiation field.

Figure 4 presents the optimal P-i-N HOT photodiode structure with a very low doping concentration of approximately 10^13^ cm^−3^ in the active i-region. The wide gap regions surrounding the absorber (cap P-type contact layer and N-type buffer) act as barriers, reducing the dark current in the absorber and not contributing to it themselves, and they prevent a tunnelling current under reverse bias. The i-active region can be completely depleted depending on the doping concentration and reverse voltage. At the doping range of approximately 10^15^ cm^−3^ (which includes Rule 07), the 5-μm-thick MWIR absorber is completely depleted by applying a relatively high reverse voltage between 10 V and 30 V—see Figure 4b. Similar relationships exist for LWIR photodiodes. However, for the doping range recently achieved in Teledyne (from 5 × 10^13^ to 1 × 10^13^ cm^−3^), the fully depleted i-region is reached for a reverse bias from 0.4 up to zero volts, respectively. This range of reverse polarization of photodiodes is typically used to read the signal through the ROIC in the array pixel.

For a fully depleted P-i-N photodiode, the dark current density can be calculated from the expression
(3)$$J_{dep}=\frac{qn_{i}t_{dep}}{\tau_{SRH}}$$
where *t_dep_* is the depletion region width. The depletion dark current exhibits a temperature dependence given by *n_i_*.

Figure 5 summarizes the dark current densities of selected IR detectors operating at 300 K. In addition, the curves defined by Rule 07, Law 19 and Rule 22 are marked for comparison purposes. The curve marked “substrate-off” corresponds only to the background radiation limit, while “substrate-on” is also due to the device design. The “substrate-on” radiation limit is increased by the square of the substrate refractive index—for CdZnTe, it is a factor of 7.3. The total background radiation incident on the detector is increased due to the reflection of radiation (emitted from the neutral regions of the p-and-N photodiode) from the substrate–air interface [12,13].

As shown in Figure 5, Teledyne and Lynred have fabricated fully depleted photodiodes that operate at the “substrate-on” radiative limit in the MWIR spectral range. The removal of the substrate provides a further opportunity to improve the detector performance. The BLIP performance has the greatest impact on the current density for photodiodes operating in the LWIR region, while, for the SWIR region, the advantage in using fully depleted devices is very small from a dark current perspective.

Figure 6 collects the spectral detectivities published in the literature for different groups of single-element IR photodetectors operating at room temperature, including those fabricated from low-dimensional solids such as 2D materials. The performance of standard photodetectors made of HgCdTe, T2SLs, lead chalcogenides (PbS and PbSe) and InGaAs is sub-BLIP. It is interesting to note that the performance of T2SL-based interband quantum cascade infrared photodetectors (IB QCIPs) is comparable to HgCdTe. As shown in Section 2, the adoption of the α/*G_th_* standard indicates theoretically the superior performance of HgCdTe photodiodes compared to 2D material photodetectors. However, the literature data collected in Figure 6 for 2D photodetectors contradict this conclusion, with more considerations given in Section 5.

Figure 7 shows the current performance of HOT photodetectors and at the same time indicates future possible further development. At the current stage of HgCdTe technology development, the semi-empirical Rule 07 does not meet the original assumptions [12,43]. At this point, it should be noted that this metric is closely related to the p-on-n photodiode limited by the Auger 1 mechanism due to an extrinsic doping concentration in the active region of ~10^15^ cm^−3^. Figure 7 shows that, according to Law 19, the detectivity of P-i-N HgCdTe photodiodes with doping at the level of 1 × 10^13^ cm^−3^, operating at room temperature in the spectral range above 3 μm, is limited by background radiation (with a *D^*^* level above 10^10^ Jones) and can be improved by more than one order of magnitude compared to Rule 07 predictions in the LWIR spectral region. Among the material systems used in HOT LWIR photodetector fabrication, only HgCdTe meets the expectations required to achieve the BLIP conditions: low doping at the level of 10^13^ cm^−3^ and a high SRH carrier lifetime in the 1 ms range.

## 4. Interband Quantum Cascade Infrared Photodetectors (IB QCIPs)

As is commonly known, the responsivity of conventional photodiodes is closely related to the diffusion length, and increasing the absorber thickness well beyond the diffusion length no longer gives the desired improvement in the signal-to-noise (*S*/*N*) ratio. In other words, when the absorption depth is longer than the diffusion length, as for HgCdTe photodiodes operating in the LWIR spectral range, only a limited fraction of the photogenerated charge carriers contribute to the quantum efficiency (*QE*). Especially at high temperatures, where the diffusion length is usually reduced, this effect is most pronounced.

To circumvent the limitations of the reduced diffusion length, new detector designs such as intersubband (IS) ambipolar quantum cascade photodetectors (IS QCIPs), derived from quantum well infrared photodetectors (QWIP) and quantum cascade lasers (QCL), were introduced in the early 2000s to effectively increase the absorption efficiency [47]. The distinguishing feature of this type of photodetector is that they can be implemented in chemically stable material systems with well-established epitaxial growth, such as molecular beam epitaxy (MBE), and device processing technologies. Most IS transition-based devices are fabricated based on A^III^B^V^ materials: GaAs QWs with AlGaAs-matched barriers or InGaAs QWs with AlInAs-matched barriers. They are also characterized by better uniformity and a lower surface leakage current.

However, in the last decade, it has been shown that interband (IB) unipolar QCIPs based on type-II A^III^B^V^ superlattices are more effective devices. The performance of IB QCIPs is higher than that of IS QCIP ones, mainly due to the much longer carrier lifetime, which translates into two orders of magnitude lower saturation current densities for IB QCIPs than reached for IS QCIPs—see Figure 8.

Figure 9 compares the estimated peak detectivities limited by Johnson noise (based on the measured *R*_0_*A* product and responsivity at zero bias) for both types of QCIPs with commercially available HgCdTe photovoltaic detectors. It is clearly shown that the performance of IB QCIPs is comparable to that of HgCdTe. An additional advantage of QCIPs is that, thanks to the strong covalent bonds of A^III^B^V^ semiconductors, they can operate at temperatures up to 350 °C, which is not possible for the HgCdTe counterpart. However, the sophisticated structures of IB QCPs, with many interfaces and strained thin layers, are the main problem in technology development, also due to increased production costs. Thus far, attempts to obtain a high-quality infrared detector array with IB QCIPs pixels are not very optimistic. The minimum noise equivalent difference temperature (*NEDT*) of 28 mK at a long integration time (up to 30 ms) was obtained at 120 K with f/2.3 optics using a 320 × 240 focal plane array (pixel size of 24 × 24 μm^2^) and standard commercial read-out circuit technology [49]. In the longer term, however, the flexibility of structure designs and material parameters provides tremendous scope for improvements in their performance.

## 5. Two-Dimensional Material Infrared Photodetectors

Two-dimensional materials are promising candidates for high-performance photodetectors operating over a wide spectral range, due to novel and unusual properties such as the following:the electronic states, with a typical thickness of less than 10 nm, are easily tuned by external fields (e.g., ferroelectric field, gate-induced electrostatic field and photogating localized field);their bandgaps, ranging from 0 eV for graphene up to 6 eV for hexagonal boron nitride (h-BN), make it possible to produce photodetectors that operate from ultraviolet (UV) up to far IR or even terahertz (THz);the bandgaps are closely related to the number of layers—increasing the number of layers, the band gap is reduced;for some TMDs, such as MoS_2_, MoSe_2_, WS_2_ and WSe_2_, monolayers are direct bandgap semiconductors, while the bulk materials are indirect bandgap semiconductors.

Due to the fact that there is no need to pay attention to the lattice matching, as in conventional semiconductors, the random stacking and fabrication of 2D material devices is a tremendous advantage. A stack of atomic planes placed on top of each other, held together by vdW forces, leaves no dangling bonds, which promotes the fabrication of vertical heterostructure devices. Additionally, the weak vdW interactions allow the development of 2D materials on a large scale, regardless of the substrate, and the integration of 2D materials into silicon chips, which represents huge potential in the production of electronic and optoelectronic devices.

At the present stage of technology, however, the following issues influence the development of 2D material photodetectors:they are susceptible to absorbates owing to their atomic thicknesses, defects and doping;they lag significantly behind the traditional infrared material technologies, including low-quality heterojunctions between the monolayers of vertically stacked TMDs;they do not show a distinct advantage over the conventional 3D bulk materials and, in addition, the carrier mobility varies along different crystal orientations (in consequence, the design of devices along the preferred direction is crucial for high responsivity and a short time constant);their performance, in the majority of cases, is dictated by the nature of layer stacking (twist, spacing, etc.) and their environment (strain, pressure, etc.);despite the large absorption coefficient, the atomically thin materials are not applicable for intense light, which leads to a poor linear dynamic range.

Despite the mentioned drawbacks, a new class of materials (2D materials and quantum dots/nanowires) can compete with standard commercial photodetectors. Many papers have been published on this topic; some of them are overly enthusiastic.

Figure 5 shows that the estimated dark current densities for 2D material photodetectors selected from the literature represent the record/lowest values to date for photodetectors operating at room temperature. Figure 6 and Figure 7, on the other hand, show published higher detectivities of 2D layered photodetectors compared to commercially available standard detectors, including photodiodes with HgCdTe. The available literature describes photodetector performance that exceeds the physical limit—limited by the influence of background radiation. This is best seen in Figure 5 and Figure 10. This second figure collects the experimental dark current densities, versus the reverse of the *λ_c_T* product, of a wide group of photodetectors operating at 300 K. The red curve is theoretically calculated according to the Law 19 metric, i.e., it determines the dark current density resulting from the influence of background radiation at the 2π field of view (FOV) and 300 K. The highlighted values in Figure 10 were obtained from texts, tables or graphs included in the available publications. For photodetectors made of 2D materials, the dark current density is estimated based on available data in published papers.

The authors of several published papers have underlined cases where the performance of 2D layered photodetectors is overestimated due to the lack of precisely defined procedures for their characterization [57,58,59,60]. As is shown in a recently published paper [60], the misinterpretation of layered photodetector performance is due to

incorrect noise estimates;the miscalculation of the device active area and radiation power density;the contradictory bandwidth of measured responsivity and noise.

Both shot and generation-recombination (GR) noises depend on the photoconductive gain, *g*. The implementation of an incorrect shot noise expression ($I_{sh}=\sqrt{2qI∆f}$) instead of $I_{sh}=\sqrt{2qIg∆f}$) leads to a false increase in the signal to noise (*SNR*) ratio by a factor of $\sqrt{g}$. The last equation shows that the shot noise calculation error rises for higher gains and is remarkably important for high-internal-gain photodetectors.

The development of a low-dimensional solid (LDS) photodetector requires proper characterization procedures consistent with those used for typical bulk-based photodetectors.

The highest-performance 2D material photodetectors are among phototransistors. Figure 11 explains how a phototransistor operates resembling a simple photoconductor, where the signal is caused by the generation of an electron–hole pair when one type of carrier is trapped by localized states (nanoparticles and defects). Typically, the photoconductive gain may be simply calculated by the ratio of the carrier lifetime to the transit time between the detector’s contacts. When the carrier’s drift length is greater than the contact distance, the charge swept from one electrode is instantly replaced by the injection of an equal free charge on the second contact, making the free charge circulate in the circuit until recombination, leading to signal amplification—a photoelectric gain. The phototransistor’s active region is separated from the substrate by an insulator, allowing the application of a gate voltage, *V_G_*, to control carrier transport in the active region. The active region was found to be more susceptible to the local electric field than typical bulk materials, and the optical generation can strongly modulate the conductivity of the channel by the external gate voltage, *V_G_*. Under such conditions, a much higher optical gain can be achieved. Generally, however, the gain in phototransistors can vary significantly versus the gate voltages due to the gate-controlled number of trap states.

Below, we will discuss the difference in dynamic behavior between conventional and layered IR photodetectors.

Practical applications require high-performance photodetectors characterized by a wide and linear dynamic range of operation, meaning that the photocurrent exhibits a linear dependence on the incident radiation power before absorption saturation. In this case, *I_ph_ α P^α^*, where *α* is close to 1. However, in the case of LDS photodetectors, including 2D material photodetectors, a nonunity exponent, 0 < *α* < 1, is often found, which is due to complicated carrier generation-recombination and trapping mechanisms. The responsivity is determined from the formula *R = I_ph_/P*, so *R α P^−^*^(*1−α*)^. Figure 12a shows the net photocurrent and responsivity nonlinear radiation power dependence. If level of the incident radiation increases, the carriers are gradually captured, leading to complete trap filling and a decrease in the carrier lifetime and photoelectric gain. The gain in a photoconductor is proportional to the lifetime, while the response bandwidth is inversely proportional to the carrier lifetime. It follows that the gain × bandwidth product of the photoconductor is limited—see Figure 12b.

In the low-power range of incident radiation, the sensitivity is not influenced due to the high trap state density. In general, however, the sensitivity values published in the literature are not a criterion for the comparison of the quality of photodetectors, as they are measured for different incident radiation power densities. It has been assumed that to determine the current responsivity of photodetectors, it is common to use an incident radiation power density that is several orders of magnitude <1 mW/cm^2^ [61,62].

Figure 13 summarizes [28,29,33,35,46,63,64,65,66,67,68] the responsivity of selected LDS devices, including 2D material photodetectors, with current responsivity reaching ~10^10^ A/W and ultrahigh gain >10^9^ electrons per photon, for the wavelength <2 μm. The InGaAs and HgCdTe photodiodes’ photoelectrical gain = 1, due to the separation of minority carriers by the electric field of the depletion region. For LDS photodetectors, the high-sensitivity detectors have a low response rate, being independent of the spectral range and being confirmed by the experimental data presented in Figure 14 (from visible (VIS), through SWIR and MWIR to LWIR). Response times of up to a few seconds are observed for LDS photodetectors in the visible range. This is a challenge in reaching both high photoresponsivity and a fast response simultaneously.

## 6. Colloidal Quantum Dot Infrared Photodetectors

Quantum dot (QD) photodetector development (with origins in the 1990s) has seen an evolution from epitaxially grown self-assembled QD detectors to a new generation of colloidal nanocrystal-based devices. A particular development in the latter has been observed in the last decade, where an active region is based on 3D semiconductor nanoparticles. CQDs offer a potential alternative to the InGaAs, InSb, InAsSb and HgCdTe as well as T2SLs in the SWIR and MWIR spectral regions. They are typically fabricated by conducting polymer/nanocrystal blends or nanocomposites. Nanocomposites are characterised by narrow bandgap energy, e.g., A^II^B^VI^ compounds (HgTe, HgSe) and A^IV^B^VI^ compounds (PbS or PbSe). Table 2 summarizes the pros and cons of CQD devices.

Early studies of CQD-based detectors were reported for the NIR to MWIR photoconductors. Despite simplicity in device architectures, their performance was restricted by the dark currents, *1/f* noise and problems in doping concentration monitoring.

The responsivity of hybrid photodetectors (in fact, phototransistors—see Figure 11) can be considerably higher than commercially available detectors; however, a significant decrease in response time (bandwidth) is visible. The majority of these detectors exhibit a reduced linear dynamic range conditioned by the charge relaxation time (see Figure 12), leading to a decrease in responsivity versus radiation power. Figure 15 presents the current responsivity versus radiation intensity for a hybrid PbS/graphene QD photodetector with selected QD sizes [80]. This figure also shows the responsivity dependence on the QD size. For small QDs (*d* = 2.65 nm), the responsivity reaches >10^7^ A/W for low radiation power. Typically, the response time (millisecond range and longer) is three orders of magnitude longer than commercially available detectors (microsecond range and shorter). The recovery process exhibits a slow component > 1 s connected with traps in the QDs [46]. The presented drawbacks restrict potential applications. The high current responsivity of the hybrid detectors is determined by the high photoelectric gain. As this gain also increases shot noise and g-r noise, this does not usually translate into higher detectivity.

One other point should be borne in mind in the context of FPA fabrication. The hybrid detector needs a three contact device (like a phototransistor), affecting the FPA’s power consumption and limiting the fill factor. Moreover, the hybrid pixels’ complicated design influences FPAs’ uniformity and operability, and QDs’ nonuniformity considerably restricts the performance [81].

Current research is directed towards CQD photovoltaic devices with lower *1/f* noise and dark currents. Thus, our attention will be focused towards photodiodes with more sophisticated structures.

The CQD-based photodiodes’ current responsivity was reported within the range of 100 mA/W-1 A/W, which corresponds to external quantum efficiency ~10–80%, being higher in the short-wavelength infrared region (SWIR) (refer to Figure 16). These external quantum efficiencies (EQEs) are significantly higher than reached for epitaxial (self-organising) QD photodetectors—typically reported at ~2%. Typical QEs for commercially available InGaAs, InSb, HgCdTe and T2SL-based detectors are presented in Figure 16.

The CQD layers are reported to be amorphous, allowing integration with ROIC substrates, as presented in Figure 17, with no pixel or array size limitations and a simple fabrication procedure. The CQD-based detectors’ monolithic integration into ROIC does not demand any hybridization process. The single pixels are defined by the ROIC’s metal pads. To synthesize colloidal nanocrystals, the compounds are injected into a flask and the required shape and size are reached by the reagent concentrations’ monitoring, ligand selection and temperature. This top-surface photodetector offers a 100% fill factor and is compatible with postprocessing CMOS electronics.

The most common photodiode stack accompanied by bandgap diagrams is presented in Figure 18a. The structure with a QD absorber contains additional thin-film transport layers made of metal oxides, organics (or 2D materials) monolithically integrated on CMOS read-out integrated circuits. The internal electric field directs electrons and holes in opposite directions through the transport layers. As shown, photodiode structures are fabricated in either homojunction form (top diagram) or heterojunction form (bottom diagram).

Increasing the thickness of the CQD layers, and at the same time an increase in the radiation absorption, is practically challenging due to the high likelihood of crack formation in the CQD film and the limited diffusion length of photoexcited carriers in CQD layers. A short diffusion length is a consequence of the low mobility of the carriers (several cm^2^/Vs in PbS QDs and below 1 cm^2^/Vs in HgTe QDs). Therefore, many recent efforts have focused on integrating CQD thin films with resonant optical structures to enhance light absorption. Photonic structures such as back reflectors, gold gratings and simplified Helmholtz resonators have been shown to confine and concentrate incident electromagnetic fields, thereby increasing the EQE of the CQD-based devices. Figure 18b presents an example of the pixel cross-section of a HgTe CQD FPA with the thickness of the absorption layer around 400 nm [56]. The pixel contains a pair of electrodes, the pixel electrode, which acts as both an electrical contact and as a back reflector and the ground electrode. The detector is biased by the pixel and ground electrodes.

In the fabrication of CQD photodetectors, the post-synthesis ligand exchange process is perhaps the most critical step in determining the detector’s ultimate performance. Ligands are molecules bound on the surface of a CQD. Due to the high surface-to-volume ratio, the surface has a strong influence on the CQD’s physical properties and surface ligands play an important role in the CQD’s stability and electrical properties. The aim of the ligand exchange process is to transfer the QDs to suitable ligands and solvents, control the majority carrier type and tune/increase the charge mobility. To reach efficient carrier transport, isolating long organic ligands should be replaced by small and/or ionic ligands. Efficient ligand exchange routes have been presented in the literature [31,88,89,90,91].

The lead chalcogenide family is mostly represented by lead sulfide (PbS) CQDs operating in the 1–3 μm NIR spectral range [92]. PbS CQD vertical technology, including the solution process, is compatible with the CMOS read-out technology used in monolithic FPA fabrication. The 100-nm-thick PbS CQD stacks may be implemented as effective active layers [93]. The conventional hybrid FPAs are normally restricted to 1 megapixel range arrays due to the small detector wafer and low throughput. In this configuration, a 5 μm pixel pitch was reported [3]. With a thin-film active layer integrated monolithically with ROIC, a <1 μm submicron pixel size can be fabricated (Figure 19). Currently, the CMOS VIS sensor state-of-the-art is 0.9 μm [3].

Figure 20a shows the absorption peak’s dependence on the nanocrystal size, where, for a 5.5 nm diameter, the peak absorption occurs at 1440 nm, and for 3.4 nm, the peak could be reached at 980 nm, respectively. This size-dependent flexibility can be implemented in hyperspectral sensors. The SWIR PbS CQD reached detectivities >10^12^ Jones (see Figure 20b) and QEs comparable to commercial 300 K InGaAs photodiodes. Vafaie et al. [88] reported on a high-concentration bromine processing and passivation step to enhance the responsivity of a 1550 nm PbS. The photodiode exhibited 80% EQE and a 10 ns response time, being the lowest reported fall time of an SWIR device fabricated by solution-processed materials.

HgTe CQDs exhibit a spectral response within the SWIR-THz ranges. The first BLIP HgTe CQD photovoltaic detector operating in the MWIR (*λ_c_* = 5.25 μm at 90 K) was demonstrated in 2015 by Guyot-Sionnest et al. [95]. The low carrier mobility in MWIR CQD films hinders the detector’s performance. This drawback could be circumvented by a proper CQD chemical treatment. Further progress in the sensitivity and time response was reached by Ag_2_Te heavily p-type doping [96]. Figure 21a shows the HgTe CQD photovoltaic detector design on an indium tin oxide (ITO)/sapphire substrate. The ~400 nm HgTe CQDs were deposited layer-by-layer by drop-casting. A solution of ~10 nm Ag_2_Te nanoparticles was then spun over the HgTe layer to create a strongly p-doped HgTe CQD top layer. A two/three times increase in *D** was reached for the detector supported by a plasmonic nano-disc in comparison with a device without an absorption enhancement.

More recently, gradient homojunction HgTe CQD photodiodes (see Figure 21b), using improved ligand exchange methods, have achieved even better rectification behavior via gradual doping level changes, leading to an increase in room-temperature detectivity to *D** > 10^9^ Jones for a cut-off wavelength close to 4 μm [31]. Figure 21c shows the 300 K photoresponse spectra of CQD p-i-n HgTe homojunctions with *λ_c_* equal to 1.5 μm, 1.9 μm, 2.5 μm and 3.6 μm, being related to the absorption’s dependence on the QD size (see Figure 21d).

Figure 22 collects the highest published detectivities for selected photodetectors (PC-PbS, PC-PbSe, PV-Si, PV-InGaAs, PV-HgCdTe and CQD IR photodiodes) operating at 300 K. The presented data indicate sub-BLIP performance. The performance trend lines for HgCdTe photodiodes, IB QCIPs and 2D material photodetectors are also presented based on the experimental data collected in Figure 6. The detectivity of the CQD photodetectors in the MWIR region is comparable to that of HgCdTe photodiodes, indicating significant progress in CQD development. This progress is particularly evident in the SWIR, where the CQD-based detectors’ performance is even higher; however, the CQD photodiodes’ high detectivities in the SWIR and MWIR do not correlate with the high dark current densities marked in Figure 10. As presented in this figure, the current densities of CQD photodiodes are higher than those observed for SWIR HgCdTe photodiodes. These contradictory data are probably due to errors made in the measurements of the detector parameters, as pointed out in several papers [57,58,59,60].

The majority of the CQD detector research has been directed towards the single-pixel devices but CQDs offer also potential as an alternative in the traditional image sensor field, advancing from Si compatibility, solution processability and simplified device fabrication [94,97]. To date, several studies of SWIR and MWIR imagers have been reported. It is predicted to extend the spectral ranges to longer wavelengths. Table 3 summarizes key figures of merit for different monolithic CQD image sensors. This monolithic manufacturing provides affordability for many applications that could not consider IR data collection, such as smart agriculture, medical, surveillance and consumer applications and the automotive sector.

## 7. Conclusions

Currently, microbolometer arrays are the most widely used for uncooled IR detectors; however, these thermal detectors exhibit modern sensitivity and limited response times—usually in the millisecond range—and are not very useful for applications requiring multispectral detection. Bolometric detection typically cannot compete with the sensitivity or time response of semiconductor-based photon detectors. To circumvent this limitation, further efforts have been directed towards room-temperature photon detectors with better sensitivity and shorter response times.

Despite long-standing concerns regarding HgCdTe’s uniformity and toxicity [99], this compound occupies the main position among HOT photodetectors, especially in the LWIR range. For more than 60 years, HgCdTe has successfully competed with alternative technologies to include extrinsic silicon devices, lead–tin telluride devices, Schottky barriers on silicon, AlGaAs multiple quantum wells, T2SL and especially silicon microbolometers. Lately, important progress has been demonstrated in the fabrication of IR LDS detectors (2D materials and particularly CQDs) to include possible integration into imaging cameras. Both technologies have become exciting frontiers in the MWIR. It is expected that nano-fabrication and micro-electromechanical (MEMs) technology, as well as direct integration with CMOS architectures, will lead to improvements in and the widespread commercialization of FPAs containing pixels made of 2D materials and CQDs. However, given the current state of both technologies, several issues need to be addressed.

The misinterpretation of layered photodetector performance due to incorrect noise calculations, the incorrect assessment of the device’s active area and radiation power density and the contradictory bandwidth of measured responsivity and noise.The current fabrication of 2D structures, which is mostly restricted to materials exfoliated from bulk crystals, with very limited yields, scalability and reproducibility. As a result, the nonuniformity in the spectral response and response times highly deteriorates FPAs’ performance.Further improved ligand exchange methods combined with enhanced photon collection are required to increase the CQD-based photodetectors’ performance.Although the performance of IB QCIPs is comparable to that of HgCdTe photodiodes, their complex structures with multiple interfaces and strained thin layers cause problems in technology development and drive up production costs.

HgCdTe BLIP’s ultimate HOT performance limit has not been reached. To reach this limit, a ~10^13^ cm^−3^ doping concentration in the active region is required, which was recently reported by Teledyne Technologies. It has been shown that this doping level in the active region allows one to reach BLIP performance for p-i-n HgCdTe photodiodes operating above 3 μm [12,50]. It will be difficult to approach the performance presented by HgCdTe photodiodes for 2D and CQD photodetectors. This statement is supported by an analysis of the ~*α/G_t_*_h_ standard as a figure of merit for IR detectors. Teledyne has presented fully depleted 640 × 512 arrays (MWIR and LWIR operating at up to 250 and 160 K) [12,100], paving the way for the unique possibility of producing arrays with the largest format. In the next decade, HgCdTe photodiode arrays are expected to operate at 300 K in the MWIR and LWIR.

There are different methods of light coupling in a photodetector to enhance the quantum efficiency. In general, these absorption enhancement methods can be divided into four categories that use optical concentration, antireflection structures, an optical patch increase or light localization (e.g., using metasurface [101,102]). These methods can be used for photodetectors fabricated with different material systems. In our paper, we focused on the impact of the basic physical properties of different materials on the performance of photodetectors.

The future commercialization of next-generation HOT photodetectors (2D and CQD materials) will depend on their large-scale integration with existing photonic and electronic platforms, such as CMOS technologies; high operability; spatial uniformity; temporal stability; and affordability. Given this, success in the commercialization of CQD photodetectors appears to be feasible. However, there is no doubt that QD image sensors will penetrate more compact, less demanding markets and reach a high pixel density, while hybrid FPAs will retain the dominant position for high-end applications with stringent requirements.

## Figures and Tables

**Figure 1 sensors-23-07564-f001:**
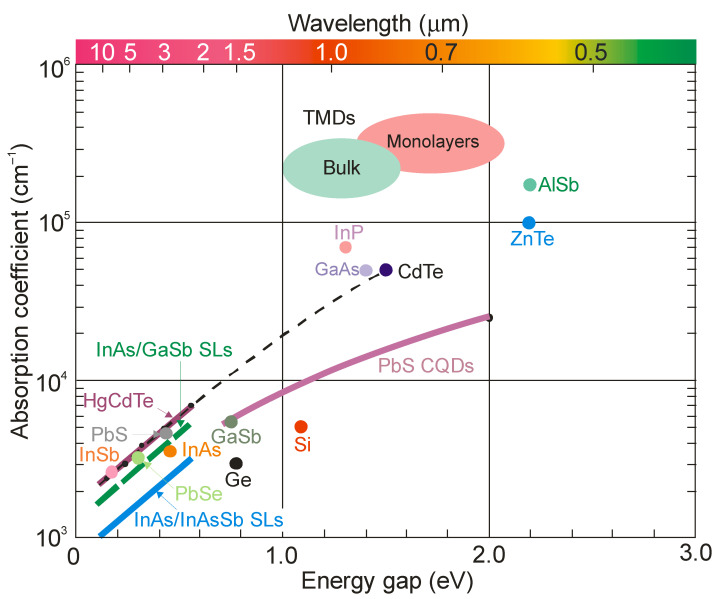
Absorption coefficient versus bandgap energy for selected semiconductor materials at room temperature. Data for HgCdTe are taken from Ref. [4]. The transition metal dichalcogenides (TMDs) (both bulk and monolayers) of Mo and W are taken after Ref. [7]. The relationship *α*(*E*) for PbS CQDs is assumed from data taken from a number of papers.

**Figure 2 sensors-23-07564-f002:**
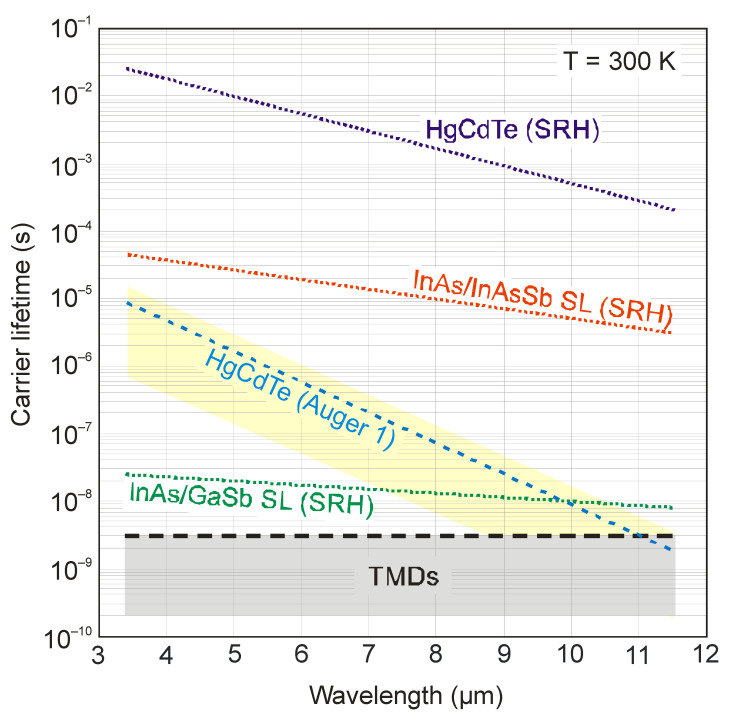
Trend lines of collected minority carrier lifetimes in four material systems: HgCdTe, InAs/GaSb T2SLs, InAs/InAsSb T2SLs and TMDs. The highest line refers to the best HgCdTe data for the lowest doping level in the active photodiode region of ~10^13^ cm^−3^ [12]. The trends for Auger mechanisms were calculated for HgCdTe and refer to the intrinsic concentration, assuming |*F*_1_*F*_2_| = 0.15. In other semiconductors, the carrier lifetime limited by Auger mechanisms may be different depending on the ratio of the effective masses and the overlap integral |*F*_1_*F*_2_|. The possible range is marked with a yellow background.

**Figure 3 sensors-23-07564-f003:**
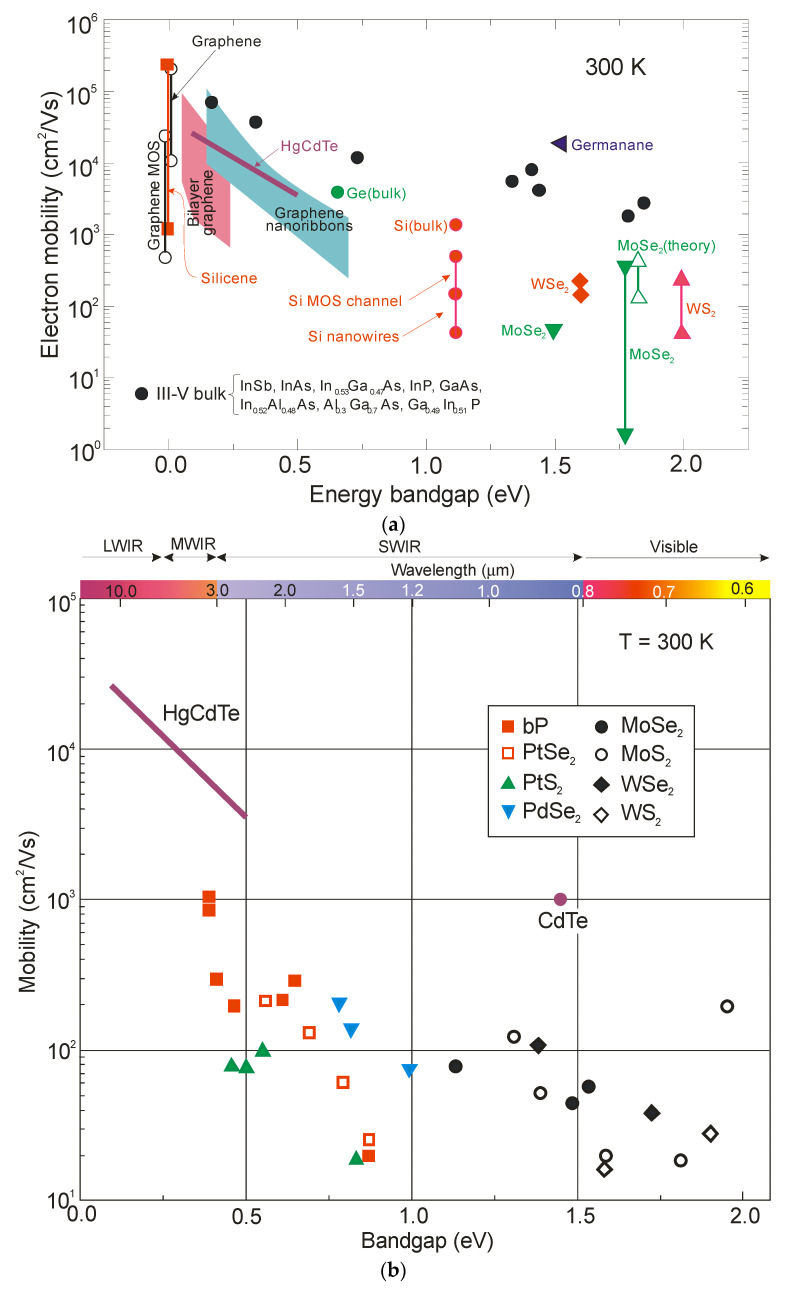
Comparison of room-temperature electron mobilities of selected layered material systems with standard semiconductors used in IR detector fabrication: (**a**) comparison with A^III^B^V^ compounds and HgCdTe ternary alloy; (**b**) electron mobility of HgCdTe (after Ref. [17]) and layer-dependent mobilities of group-6 TMDCs, bP and typical noble TMDs on back-gated SiO_2_ substrate (after Ref. [8]). Other experimental data come from various sources.

**Figure 4 sensors-23-07564-f004:**
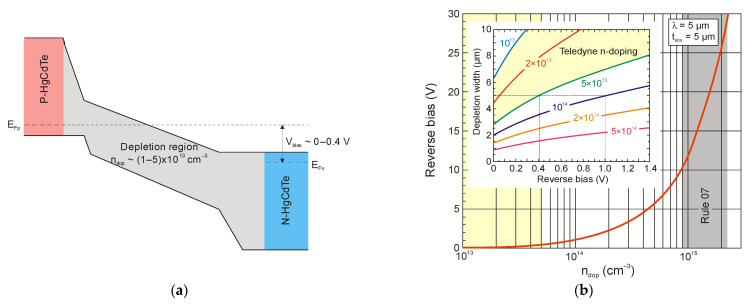
Schematic energy band diagram of a reverse-biased P-i-N HgCdTe photodiode (**a**) and calculated reverse bias voltage versus doping concentration required to deplete a 5-μm-thick MWIR HgCdTe absorber (**b**). Inset: depletion width versus reverse bias voltage calculated for selected doping concentration.

**Figure 5 sensors-23-07564-f005:**
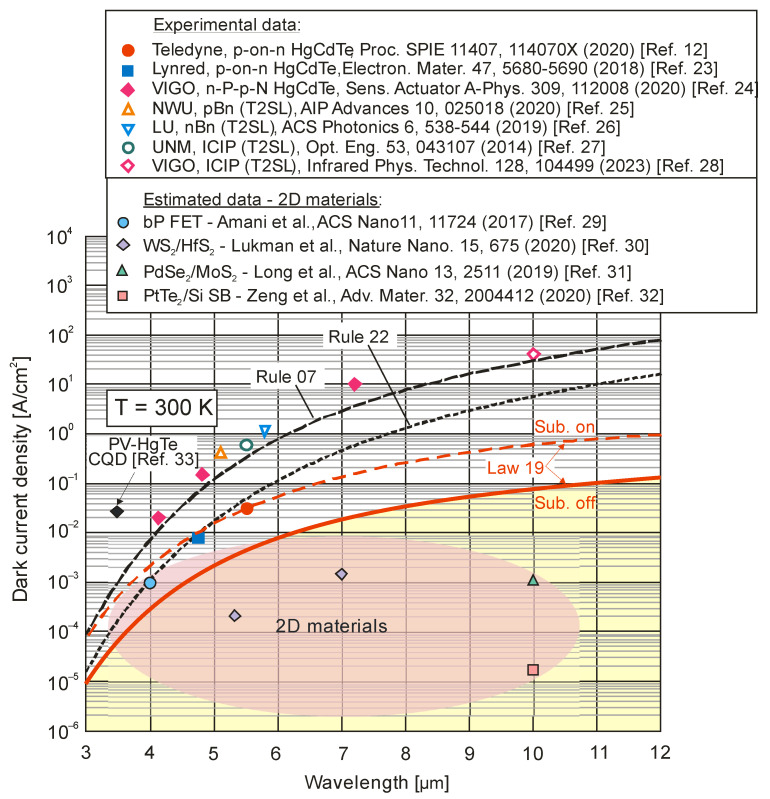
Published dark current density data for selected IR photodetectors at 300 K, compared with HgCdTe Rule 07, Rule 22 and Law 19. Experimental data are gathered for p-on-n [12,21] and N-P-p-N HgCdTe [22] photodiodes and T2SL (barrier and cascade) photodetectors [23,24,25,26]. The lowest current densities are marked for photodetectors made of 2D materials. The current density values for the latter are estimated from the data in the respective papers [27,28,29,30]. Moreover, the dark current density for an MWIR CQD HgTe photodiode is marked [31].

**Figure 6 sensors-23-07564-f006:**
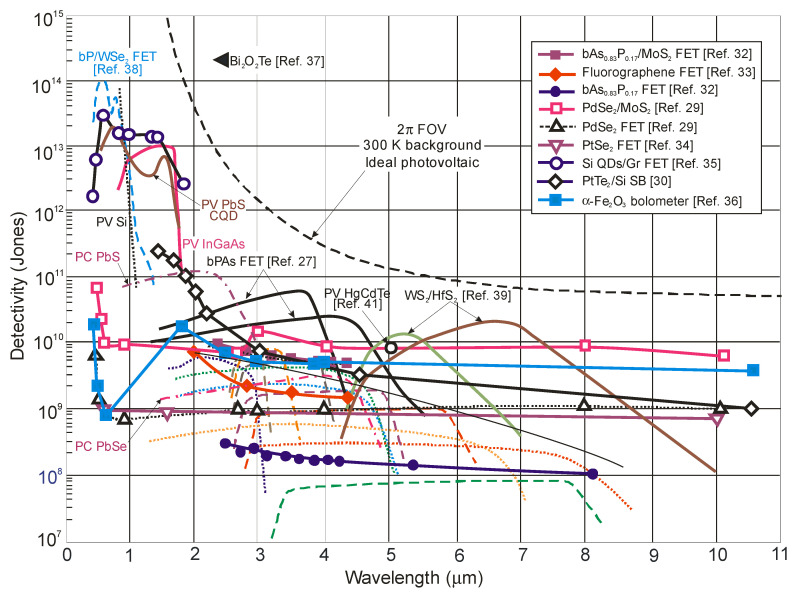
Room-temperature spectral detectivity curves for selected 2D material photodetectors [27,29,30,32,33,34,35,36,37,38,39] and the commercially available photodetectors (PV-Si and PV-InGaAs, PC-PbS and PC-PbSe, HgCdTe photodiodes (dashed lines—Ref. [40], and Ref. [41])). The spectral detectivity curves of T2SL IB QCIPs are marked by dotted lines (Ref. [42]). PC—photoconductor, PV—photodiode, FET—field effect transistor.

**Figure 7 sensors-23-07564-f007:**
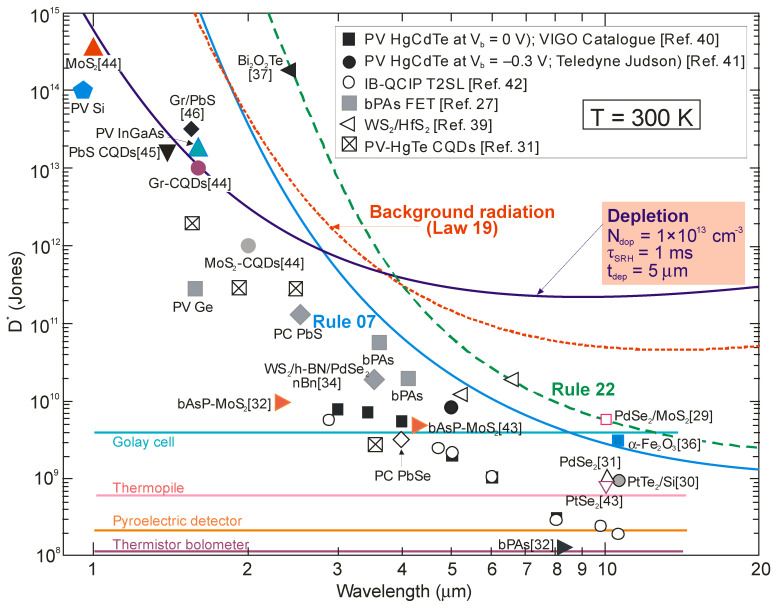
Room-temperature spectral detectivity data for commercially available IR photodetectors (PV-Si and PV-Ge, PV-InGaAs, PC-PbS and PC-PbSe, PV-HgCdTe [40,41]). Experimental data for IB QCIPs based on T2SLs [42], selected 2D materials [27,29,30,32,34,36,37,43,44,45,46] and CQD [31]. Theoretical calculations (curve labeled as “Depletion”) are for P-i-N HOT HgCdTe photodiodes assuming *τ_SRH_* = 1 ms, absorber doping level of 1 × 10^13^ cm^−3^ and active region thickness *t* = 5 μm. Also marked for comparison are the curves defined by Rule 07 and Rule 22. PC—photoconductor, PV—photodiode, FET—field effect transistors. Typical detectivity values of thermal detectors (thermistor bolometer, pyroelectric detector, thermopile and Golay cell) are also marked for comparison purposes.

**Figure 8 sensors-23-07564-f008:**
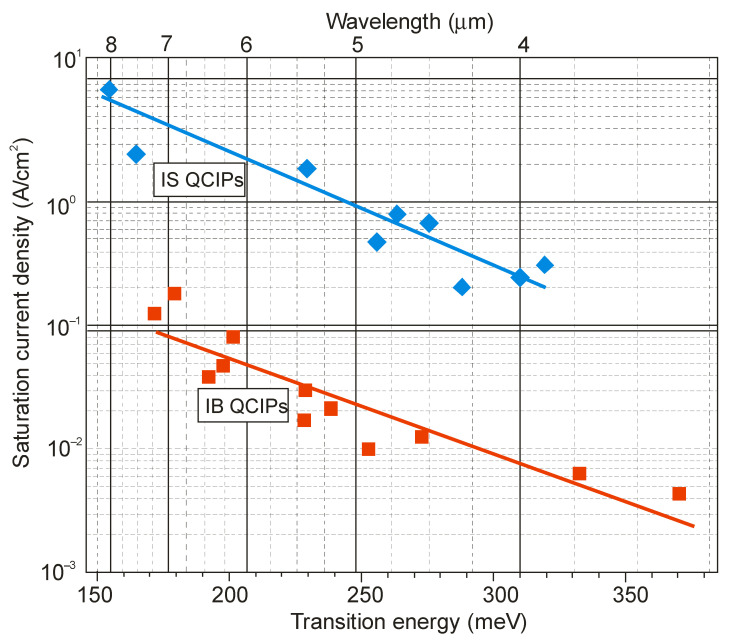
Room-temperature saturation current density of IS and IB QCIPs (adapted after Ref. [48]).

**Figure 9 sensors-23-07564-f009:**
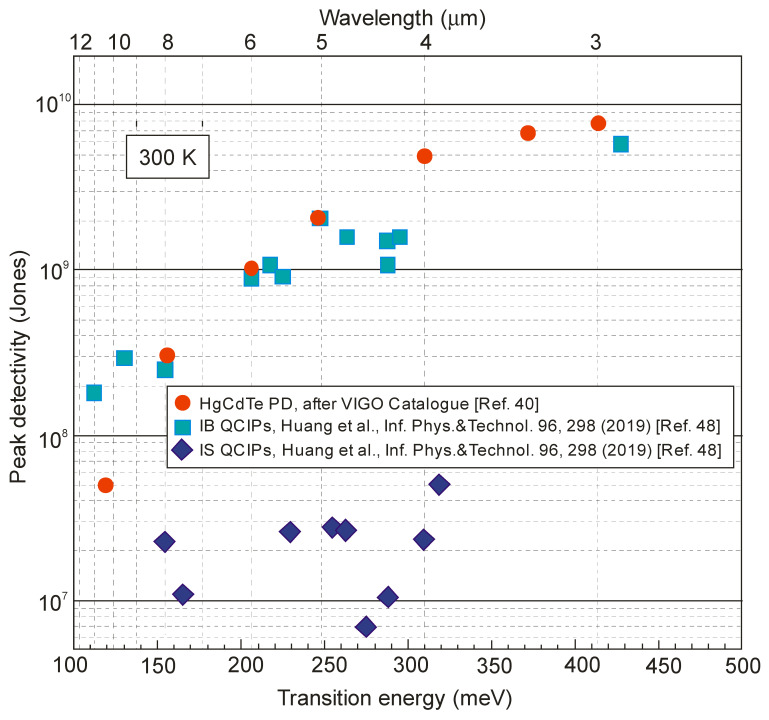
Room-temperature peak detectivity versus transition energy and wavelength for IB and IS QCIPs (Ref. [48]), compared with commercial HgCdTe photodiodes (Ref. [40]).

**Figure 10 sensors-23-07564-f010:**
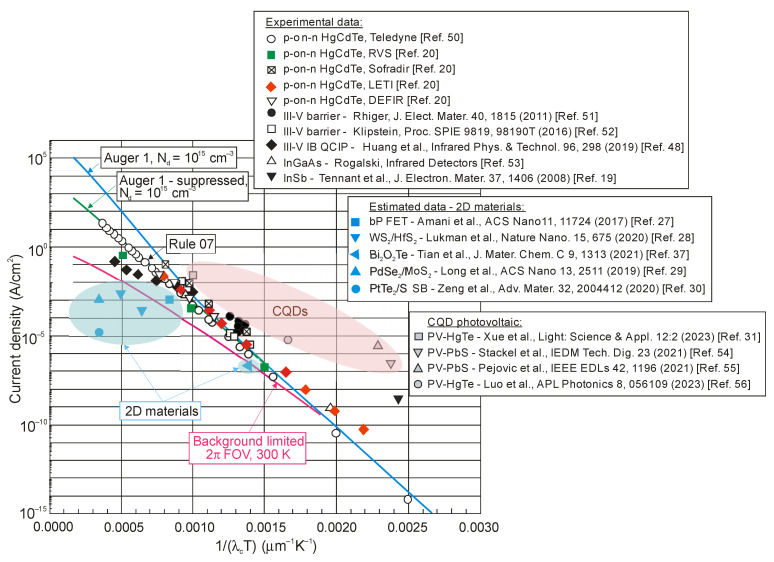
p-on-n HgCdTe photodiodes’ current density versus 1/(*λ_c_T*) product [50]. Experimental data are presented for p-on-n HgCdTe photodiodes [20,50] and alternative technologies [29,48,51,52,53] to include 2D materials [27,28,29,30,37] and CQDs [31,54,55,56]. The red curve is calculated theoretically according to Law 19 benchmark for 2π FOV and 300 K.

**Figure 11 sensors-23-07564-f011:**
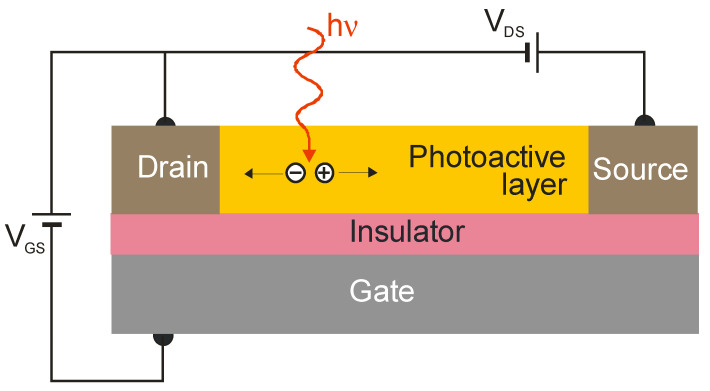
Model of the phototransistor. A current signal is usually converted to a current mode amplifier.

**Figure 12 sensors-23-07564-f012:**
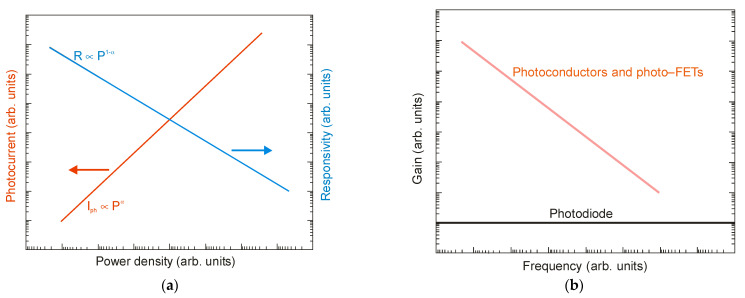
Dynamic characteristics of photodetectors: (**a**) net photocurrent and responsivity radiation power dependence; (**b**) relationship between gain and frequency.

**Figure 13 sensors-23-07564-f013:**
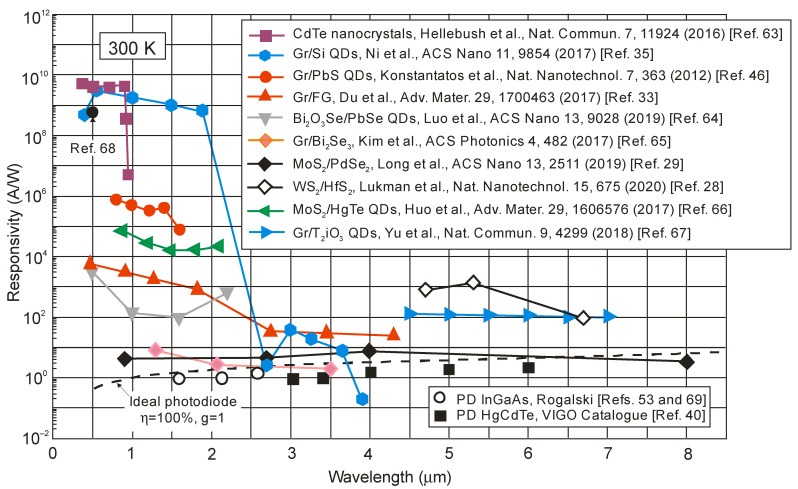
Current responsivities of LDS IR photodetectors at 300 K [28,29,33,35,46,63,64,65,66,67,68]. The performance of HgCdTe [40] and InGaAs [53,69] commercially available photodiodes is presented along with the predicted theoretical curve.

**Figure 14 sensors-23-07564-f014:**
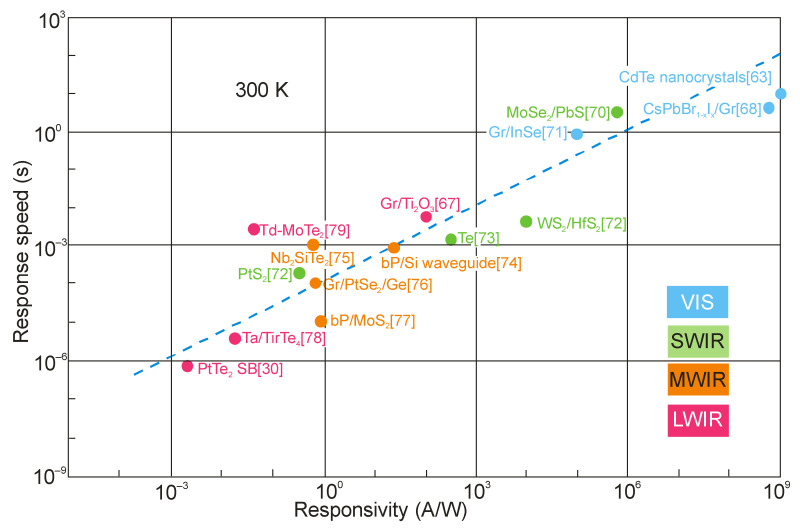
Relationship between current responsivity and response speed for 2D photodetectors operating in broad IR spectral range at 300 K. Experimental data are taken from literature as labeled [30,63,68,70,71,72,73,74,75,76,77,78,79].

**Figure 15 sensors-23-07564-f015:**
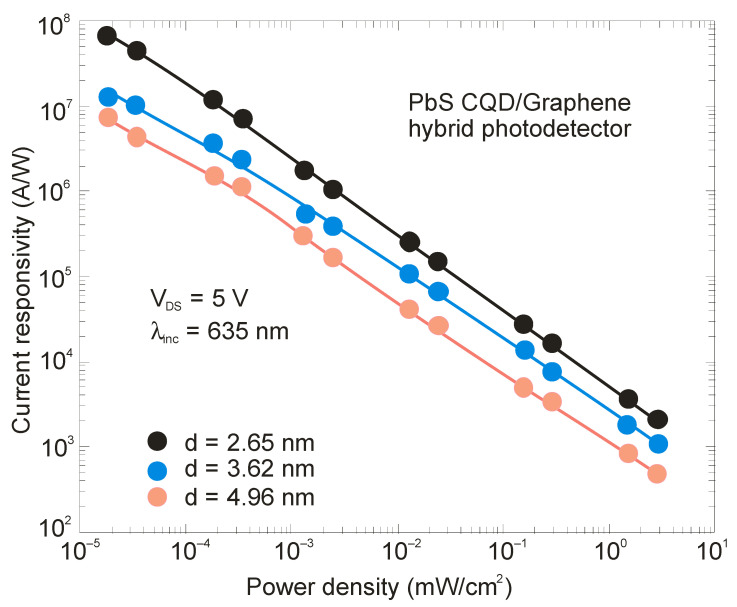
Current responsivity versus radiation power density for hybrid PbS QD/graphene device for selected QD size (adapted after Ref. [80]).

**Figure 16 sensors-23-07564-f016:**
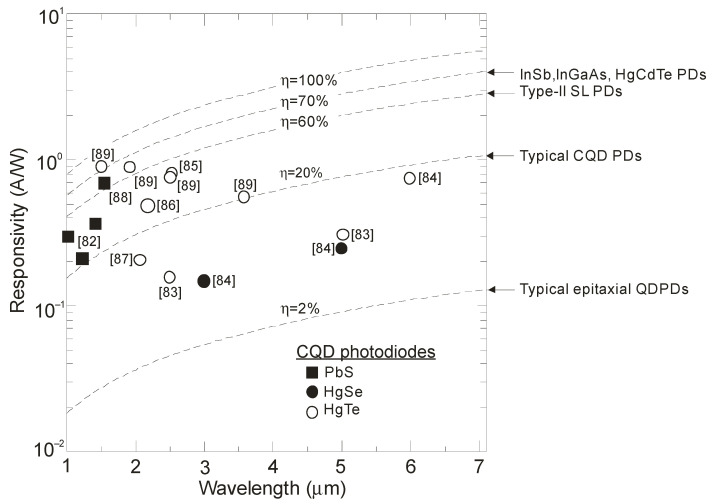
Current responsivity versus wavelength for different IR technologies at 300 K. The experimental data are extracted from Refs. [82,83,84,85,86,87,88,89]. PDs—photodiodes.

**Figure 17 sensors-23-07564-f017:**
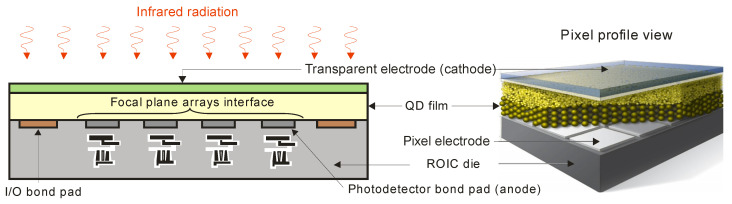
The CQD-based monolithic IR array.

**Figure 18 sensors-23-07564-f018:**
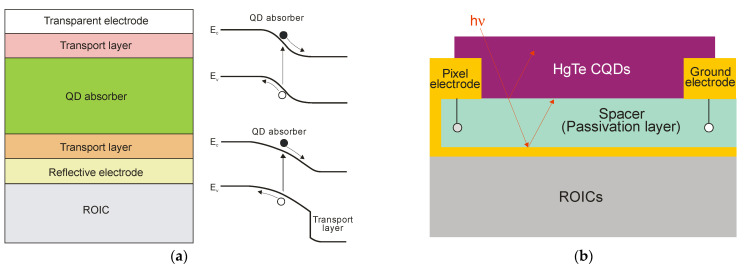
Thin-film CQD photodiodes: (**a**) the most common photodiode stacks together with bandgap diagrams for homojunction and heterojunction; (**b**) the cross-section schematic of the resonant-cavity enhanced pixel of HgTe CQD imager (after Ref. [56]).

**Figure 19 sensors-23-07564-f019:**
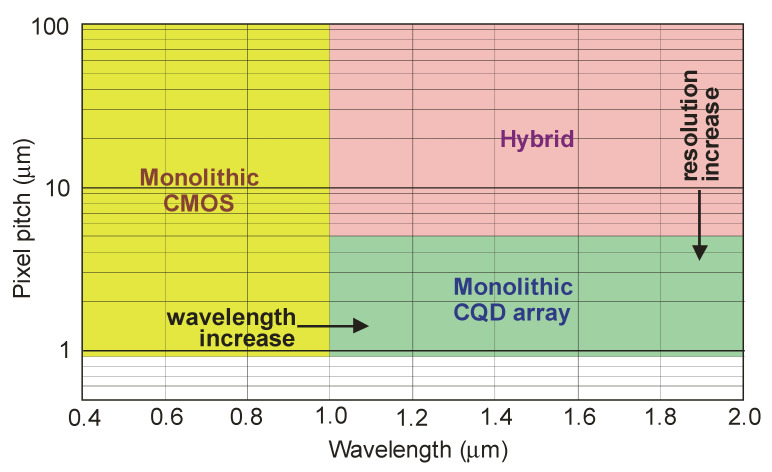
Monolithic CQD array state-of-the-art—higher resolution than hybrid alternatives and longer wavelength than monolithic Si (after Ref. [93]).

**Figure 20 sensors-23-07564-f020:**
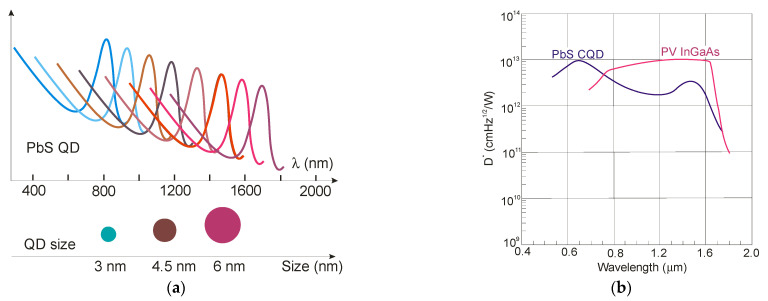
PbS CQD detector: (**a**) spectral responsivity dependence on QD size (after Ref. [93]) and (**b**) detectivity for PbS CQD and InGaAs photodiodes (after Ref. [94]).

**Figure 21 sensors-23-07564-f021:**
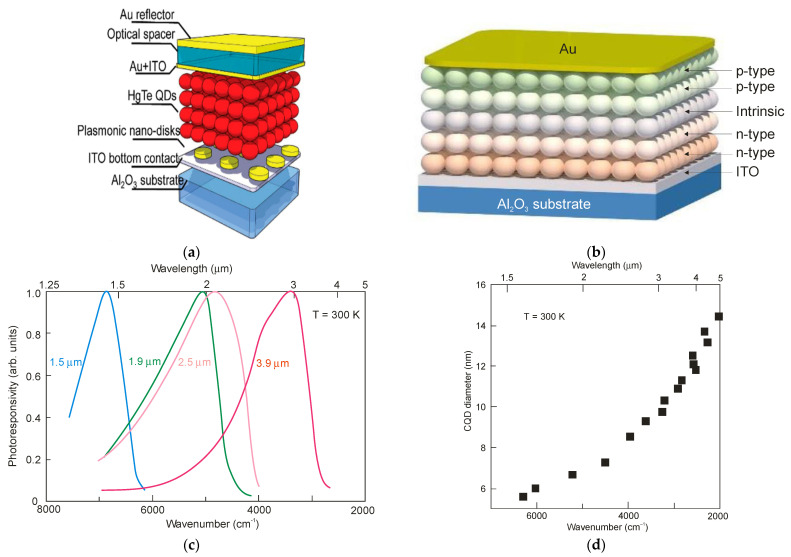
HOT HgTe CQD photodiodes: (**a**) the structure with plasmonic discs [96]; (**b**) cross-section of homojunction p-i-n photodiode [31]; (**c**) CQD p-i-n HgTe homojunction photoresponse with *λ_c_* equal to 1.5 μm, 1.9 μm, 2.5 μm and 3.6 μm [89]; (**d**) bandgap edge dependence on QD size [89].

**Figure 22 sensors-23-07564-f022:**
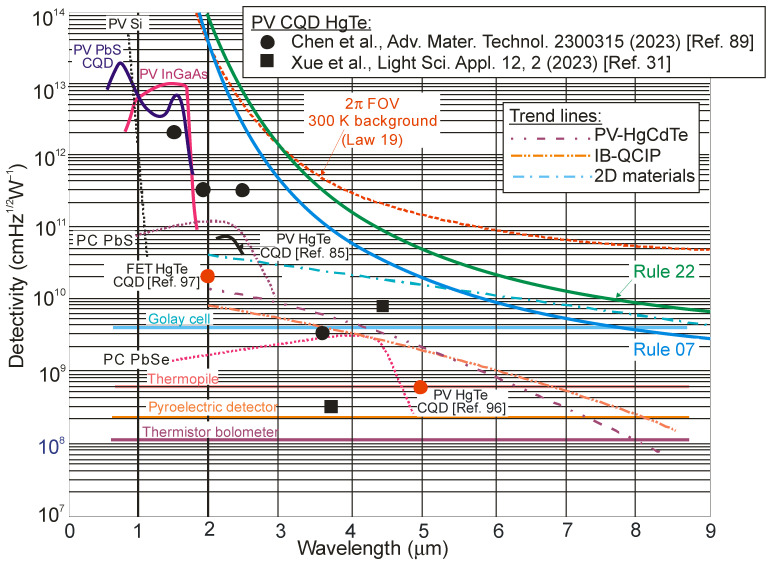
Room-temperature detectivity for CQD photodiodes [31,85,89,96,97] and commercially available photodetectors (PV-Si and InGaAs, PC-PbS and PbSe, HgCdTe photodiodes (solid lines)). For comparison reasons, the trend lines for HgCdTe photodiodes, IB-QCIPs and 2D material photodetectors are also shown based on the data presented in Figure 6. PC—photoconductor, PV—photodiode.

**Table 1 sensors-23-07564-t001:** $Estimated\sqrt{\alpha \tau }$ figures of merit for three sets of material systems at room temperature (after Ref. [3]).

Material System	Material Parameters					$\sqrt{\alpha \tau }$ [(s/cm)^1/2^]	
	Doping Concentration	Absorption Coefficient		Carrier Lifetime		MWIR	LWIR
		MWIR	LWIR	MWIR	LWIR		
HgCdTe	5 × 10^13^ cm^−3^	3.2 × 10^3^ cm^−1^	2.2 × 10^3^ cm^−1^	10 ms	0.5 ms	5.66	1.05
InAs/GaSb SLs	5 × 10^14^ cm^−3^	2.4 × 10^3^ cm^−1^	1.6 × 10^3^ cm^−1^	20 ns	10 ns	6.9 × 10^−3^	4.0 × 10^−3^
InAs/InAsSb SLs	5 × 10^14^ cm^−3^	1.2 × 10^3^ cm^−1^	8.0 × 10^2^ cm^−1^	25 μs	5 μs	1.7 × 10^−1^	6.3 × 10^−2^

**Table 2 sensors-23-07564-t002:** Pros and cons of CQD IR devices.

Pros	Cons
high QD synthesis controllability and absorption spectrum flexibility by QD size tuning, allowing uniform ensembles to be obtained; reduction in strains conditioning the epitaxial QDs—access to the spectra of active region materials; low fabrication costs (spin coating, inject printing, doctor blade, roll-to-roll printing); fabrication techniques compatible with many substrates and CMOS technologies to include direct deposition on silicon electronics with no restrictions on pixel or array size—significantly lower cost in comparison with hybrid FPAs; QD-based device to conquer less demanding markets prioritizing affordability, compactness, high pixel density (hybrid FPAs remain dominant for high-end applications with stringent requirements).	nanomaterials’ chemical and electronic passivation instability compared to epitaxial materials; QDs’ low mobility (insulating behavior—carrier transport hindered by interfaces) compared to bulk semiconductors, limiting their use in applications requiring short response times; weak chemical stability due to many interfaces with atoms exhibiting different binding; high dark currents and *1/f* noise caused by disordered granular systems; toxicity of constituent atoms (e.g., in PbS and HgTe QDs), preventing the use of CMOS foundries; monolithic FPA technology hindered by the lack of large scale, wafer-level processing.

**Table 3 sensors-23-07564-t003:** Figures of merit of CQD photovoltaic image sensors.

Parameter	PbS CQD			HgTe CQD
	IMEC Ref. [55]	SWIR V.S. Ref. [98]	STM Ref. [54]	Beijing Inst. Tech. Ref. [56]
Resolution	768 × 512	1920 × 1080	0.9 MPx	1280 × 1024
Pixel pitch (μm)	5	15	1.62	15
λpeak (nm)	1450	1470	1400	2
External QE (%)	40	15	60	14
Dark current (μA/cm2)	3.3@RT	NA	0.25@60 °C	≈6@RT
Dynamic range	82	70	54.4	NA

## Data Availability

All data generated or analyzed during this study are included in this published article.

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
