# Peer review of "Infrared HOT Photodetectors: Status and Outlook"

_sensors, 2023, doi:10.3390/s23177564_

Round 1

Reviewer 1 Report

This paper presents a comprehensive review of the infrared photodetectors operating at high temperatures. It is organized well with a good collection of technical data and results. I agree with the authors on the conclusions and I am certain that this article will be beneficial for the infrared detector community. I have several minor comments as listed below and I recommend the authors to address them as necessary.

11) Fig. 1: This figure is missing InSb, lead salts, and CQDs. Also, most of the area of this figure is used to illustrate the wavelength band outside the infrared band and most important data is squeezed into a small corner. Please consider redoing this figure to enlarge the wavelength band of interest.

22)  Line 129: It is mentioned that “It is believed that the short lifetime is due to the presence of gallium in the crystal lattice of the superlattice. For this reason, defect tolerant Ga‐free InAs/InAsSb superlattices exhibit longer lifetimes, up to several microseconds for the MWIR region.” This statement is somewhat inaccurate. InGaAs has Ga and it typically exhibits very high lifetimes. The issue is with GaSb where there is a shallow state near the valence band, which falls right within the bandgap of InAs/GaSb SLSs. In InAs/InAsSb SLS, there is no GaSb. Also, InGaAs/InAsSb SLS, which have Ga, have shown high lifetimes as high as 1 us (for MWIR).

33)  I recommend using a better reference for Rule 22, or report the empirical parameters that describes Rule 22.

44) Line 304: “quantum infrared photodetectors (QWIP)” should be “quantum well infrared photodetectors (QWIP)”.

55)  Figure 12(b): The x-axis is labeled as “Frequency”, however, the explanation in the text (Lines 443-446) indicates that it is “Power”. Also, does this plot apply for conventional photoconductors? Please clarify.

66)  Can the authors add a brief discussion on the detectivity for the hybrid detectors (Figure 15) with very high responsivity?

77)  Line 639: The statement “however, the CQDs photodiodes high detectivities in SWIR and MWIR do not correlate with the high dark current densities marked in Fig. 10” requires further clarification. If CQDs exhibit higher dark current than HgCdTe photodiodes, how is it possible for CQDs to exhibit higher D*? A brief explanation would be helpful for the readers. Is it due to discrepancy in the data reported in literature?  

Author Response

11) Fig. 1: This figure is missing InSb, lead salts, and CQDs. Also, most of the area of this figure is used to illustrate the wavelength band outside the infrared band and most important data is squeezed into a small corner. Please consider redoing this figure to enlarge the wavelength band of interest.

Response:

Additional data for InSb, lead salts, and PbS CQDc are included.

22) Line 129: It is mentioned that “It is believed that the short lifetime is due to the presence of gallium in the crystal lattice of the superlattice. For this reason, defect tolerant Ga‐free InAs/InAsSb superlattices exhibit longer lifetimes, up to several microseconds for the MWIR region.” This statement is somewhat inaccurate. InGaAs has Ga and it typically exhibits very high lifetimes. The issue is with GaSb where there is a shallow state near the valence band, which falls right within the bandgap of InAs/GaSb SLSs. In InAs/InAsSb SLS, there is no GaSb. Also, InGaAs/InAsSb SLS, which have Ga, have shown high lifetimes as high as 1 us (for MWIR).

Response:

Two sentences have been corrected:

It is believed that the short lifetime is due to the presence of a shallow state near the GaSb valence band, which falls right within the bandgap of InAs/GaSb SLSs. Ga-free InAs/InAsSb superlattices exhibit longer lifetimes, up to several microseconds for the MWIR region.

33) I recommend using a better reference for Rule 22, or report the empirical parameters that describes Rule 22.

Response:

We agree. However, in order to maintain the symmetry of the paper, we also do not provide formulas for Rule 07 and Law 19. An extended version of the paper on Rule 22 will soon appear in the Journal Electronic Materials, as noted in Ref. 20.

44) Line 304: “quantum infrared photodetectors (QWIP)” should be “quantum well infrared photodetectors (QWIP)”.

Response:

It is corrected.

55) Figure 12(b): The x-axis is labeled as “Frequency”, however, the explanation in the text (Lines 443-446) indicates that it is “Power”. Also, does this plot apply for conventional photoconductors? Please clarify.

Response:

The following explanation is introduced:

If level of the incident radiation increases, the carriers are gradually captured, leading to complete traps filling and a decrease in carrier lifetime and photoelectric gain. The gain in a photoconductor is proportional to the lifetime, while the response bandwidth is inversely proportional to the carrier lifetime. It follows that the gain×bandwidth product of the photoconductor is limited - see Fig. 12(b).

66) Can the authors add a brief discussion on the detectivity for the hybrid detectors (Figure 15) with very high responsivity?

Response:

An additional sentence is included:

The high current responsivity of the hybrid detectors is determined by the high photoelectric gain. As this gain also increases shot noise and g-r noise, this does not usually translate into higher detectivity.

77) Line 639: The statement “however, the CQDs photodiodes high detectivities in SWIR and MWIR do not correlate with the high dark current densities marked in Fig. 10” requires further clarification. If CQDs exhibit higher dark current than HgCdTe photodiodes, how is it possible for CQDs to exhibit higher D*? A brief explanation would be helpful for the readers. Is it due to discrepancy in the data reported in literature?

Response:

Unfortunately, this is the case. We have no clear explanation for these contradictory data. This is probably due to errors being made in the measurement of detector parameters, as pointed out in several papers [57-60], which is indicated in the revised manuscript.

Reviewer 2 Report

It is a complete review regarding long wavelength infrared (LWIR) detector technology developments where the tradeoff between high performance and need for cryogenic cooling. To achieve the same result at higher temperatures, photon‐based infrared detectors are utilized made from a variety of substances. Future materials directions are also elaborated.

The study can be published as long as the detection with use of metasurfaces [1,2] is discussed and connection with the present content is done.   

[1] Dual-band perfect absorber for a mid-infrared photodetector based on a dielectric metal metasurface, Photonic Research, 2021.

[2] Metasurface-enabled interference mitigation in visible light communication architectures, Journal of Optics, 2019.

[3] Metasurface Colloidal Quantum Dot Photodetectors, ACS Photonics, 2022.

Author Response

The study can be published as long as the detection with use of metasurfaces [1,2] is discussed and connection with the present content is done.

[1] Dual-band perfect absorber for a mid-infrared photodetector based on a dielectric metal metasurface, Photonic Research, 2021.

[2] Metasurface-enabled interference mitigation in visible light communication architectures, Journal of Optics, 2019.

[3] Metasurface Colloidal Quantum Dot Photodetectors, ACS Photonics, 2022.

Response:

There are different methods to light coupling in a photodetector to enhance quantum efficiency. In general, these absorption-enhancement methods can be divided into four categories that use optical concentration, antireflection structures, optical patch increase, or light localisation (e.g. using metasurface). These methods can be used for photodetectors fabricated with different material systems. In our paper, we focus on the impact of the basic physical properties of different materials on the performance of photodetectors.

In order to take into account the Reviewer’s comment, a corresponding remark on better optical coupling of the detector's active area with the incident radiation is introduced in the Conclusion.

Reviewer 3 Report

The review article titled "Infrared HOT Photodetectors: Status and Outlook" offers a comprehensive analysis of the current state and future prospects of IR photodetectors, including HgCdTe, QCIPs, 2D materials, and CQDs detectors. It effectively addresses the challenges, advancements, and potential solutions in the field, providing readers with a well-structured and informative overview. The article's content is well-researched and supported by relevant references. Given the depth of the discussions, I recommend the article for publication with minor revisions to address the following comments in order to enhance the clarity.

1. Could the authors clarify the lower threshold temperature for a condition to be considered as "high operating temperature (HOT)"? For instance, should temperatures like 90K (page 21, line 607) and 120K (page 11, line 330) still fall within the category of high operating temperatures?

2. I'm curious about the choice of plotting Figure 13 in terms of responsivity versus wavelength, as opposed to D* versus wavelength. I understand that the geometric dimensions influence the gain (responsivity) of the LDS IR detector through charge carrier transit time. However, this portion of gain that related to geometric dimensions could be cancelled out by noise. Therefore, it might be better to present D* instead of responsivity, if noise data is available.

3. A few typos have been identified in the manuscript:

On page 11, line 326, "IB QCPs" should be corrected to "IB QCIP."

On page 14, line 422, "photoresistor" should be adjusted to "phototransistor."

Author Response

  1. Could the authors clarify the lower threshold temperature for a condition to be considered as "high operating temperature (HOT)"? For instance, should temperatures like 90K (page 21, line 607) and 120K (page 11, line 330) still fall within the category of high operating temperatures?

Response:

The paper assumes that HOT detectors operate at temperatures close to room temperature - which is further indicated in the manuscript.

  1. I'm curious about the choice of plotting Figure 13 in terms of responsivity versus wavelength, as opposed to D* versus wavelength. I understand that the geometric dimensions influence the gain (responsivity) of the LDS IR detector through charge carrier transit time. However, this portion of gain that related to geometric dimensions could be cancelled out by noise. Therefore, it might be better to present D* instead of responsivity, if noise data is available.

Response:

Yes, we agree! The high current responsivity (mainly hybrid detectors) is determined by the high photoelectric gain (in SWIR region up to 109!). As this gain also increases shot noise and g-r noise, this does not usually translate into higher detectivity. The record detectivity values published in the literature are probably due to errors made in the measurement of detector parameters, as pointed in several papers, e.g. Refs 57 – 60. In many papers on low-dimensional solid photodetectors, noise values are determined theoretically and sometimes erroneously without considering the effect of photogain on the noise level.

  1. A few typos have been identified in the manuscript:

On page 11, line 326, "IB QCPs" should be corrected to "IB QCIP."

On page 14, line 422, "photoresistor" should be adjusted to "phototransistor."

Response:

Thank you! Errors corrected
